# Risk Prediction of Cardiovascular Events by Exploration of Molecular Data with Explainable Artificial Intelligence

**DOI:** 10.3390/ijms221910291

**Published:** 2021-09-24

**Authors:** Annie M. Westerlund, Johann S. Hawe, Matthias Heinig, Heribert Schunkert

**Affiliations:** 1Department of Cardiology, Deutsches Herzzentrum München, Technical University Munich, Lazarettstrasse 36, 80636 Munich, Germany; annie.westerlund@tum.de (A.M.W.); johann.hawe@tum.de (J.S.H.); 2Institute of Computational Biology, HelmholtzZentrum München, Ingolstädter Landstrasse 1, 85764 Munich, Germany; 3Department of Informatics, Technical University Munich, Boltzmannstrasse 3, 85748 Garching, Germany; 4Deutsches Zentrum für Herz- und Kreislaufforschung (DZHK), Munich Heart Alliance, Biedersteiner Strasse 29, 80802 Munich, Germany

**Keywords:** cardiovascular disease, coronary artery disease, genomics, proteomics, multi-omics, biomarkers, molecular networks, machine learning, AI, explainable artificial intelligence

## Abstract

Cardiovascular diseases (CVD) annually take almost 18 million lives worldwide. Most lethal events occur months or years after the initial presentation. Indeed, many patients experience repeated complications or require multiple interventions (recurrent events). Apart from affecting the individual, this leads to high medical costs for society. Personalized treatment strategies aiming at prediction and prevention of recurrent events rely on early diagnosis and precise prognosis. Complementing the traditional environmental and clinical risk factors, multi-omics data provide a holistic view of the patient and disease progression, enabling studies to probe novel angles in risk stratification. Specifically, predictive molecular markers allow insights into regulatory networks, pathways, and mechanisms underlying disease. Moreover, artificial intelligence (AI) represents a powerful, yet adaptive, framework able to recognize complex patterns in large-scale clinical and molecular data with the potential to improve risk prediction. Here, we review the most recent advances in risk prediction of recurrent cardiovascular events, and discuss the value of molecular data and biomarkers for understanding patient risk in a systems biology context. Finally, we introduce explainable AI which may improve clinical decision systems by making predictions transparent to the medical practitioner.

## 1. Introduction

Cardiovascular disease (CVD) is the most common source of death worldwide, ending almost 18 million lives every year [1]. In this paper, we focus on CVD caused by atherosclerosis, leading to coronary artery disease (CAD), peripheral arterial disease (PAD), and cerebrovascular disease (CeVD). Among these, CAD is the most prevalent [2]. A patient diagnosed with CAD carries plaque formations in vital vessels supplying the heart. This reduces blood flow and provokes symptoms such as chest pains and myocardial infarction (MI).

Most fatal cardiovascular events, such as stroke or MI, happen within months or years after the patient has been diagnosed with CVD [3]. In fact, many patients suffer severe complications or are in need of repeated interventions (*recurrent events*) , after their initial presentation. The events pose a great risk to the patient, and may be an economic burden to society. Being able to accurately predict and explain the risk for a patient to develop recurrent events over time is essential for current treatment protocols and is a key task for future clinical decision systems. Such decision systems would greatly facilitate medical practitioners to formulate personalized treatment plans. Specifically, the ability to explain decision patterns behind predicted risk opens up possibilities for designing novel prevention and treatment strategies, including development of drugs targeting patient-specific biomarkers.

The initial clinical presentation of a patient with CVD may offer useful information for predicting the future course of the disease. Risk prediction of CVD, as well as recurrent events, is traditionally done on the basis of clinical risk factors derived from blood tests, lab measurements, or questionnaires. A spectrum of features, or risk factors, are integrated in risk scores which are the current gold standard for risk prediction. These features include age, sex, BMI, hypertension, presence of diabetes mellitus, smoking status, family history, and cholesterol levels [2] (see Table 1, Table 2, Table 3 and Table 4).The GRACE score [4], for example, builds on risk factors such as age, systolic blood pressure, cardiac arrest at admission, and initial serum creatinine concentration to predict severe recurrent events in patients presenting with an acute coronary syndrome (ACS). Other scores, such as the REACH score [5] and SMART score [6], on the other hand, predict recurrent cardiovascular events in patients with established CVD on the basis of age, smoking status, diabetes, medical history, or carotid ultrasound findings. However, these clinical risk factors alone can only identify a subset of patients with CAD at risk for recurrent events [7]. Many recent studies therefore address this limitation in prediction performance by augmenting the set of clinical risk factors, for example by imaging data or established molecular biomarkers.

Imaging techniques such as cardiac magnetic resonance (MR) and coronary computed tomography angiography (CCTA) are noninvasive and common diagnostic tools. Features extracted from the images, for instance with image segmentation, may be used directly as phenotypes or additional risk factors to predict incident CVD or recurrent events [8,9,10,11]. Unfortunately, imaging is expensive and therefore of limited availability.

Neither clinical risk factors nor imaging data capture information about the molecular underpinnings of a disease. Indeed, knowledge of such molecular mechanisms may aid clinicians in giving a more precise diagnosis, which may ultimately allow individually tailored treatments. For example, the mRNA expression of 50 specific genes can be successfully used to classify breast cancer into prognostic subtypes [12]. To identify prognostic markers, long-term studies with individual follow-ups are necessary to retrospectively identify molecules predictive of recurrent events. Once identified, the prognostic value of the markers can be prospectively scrutinized. For diagnostic markers, it suffices to assess molecular markers upon manifestation of the disease. This may provide more accurate patient stratification, compared to current clinical practices utilizing information from lab values or images.

Genome-wide association studies (GWAS), which associate genetic variants on the DNA to a phenotypic trait, such as CVD, have become a popular and successful tool within the past decade [13,14]. This has enabled researchers to identify hundreds of heritable genetic risk loci harboring CAD-associated DNA-variants, which affect CAD via additive and non-additive effects [15,16,17,18]. Although polygenic risk scores derived from disease-associated genetic variants facilitate accurate estimates of the individual lifetime risk for incident CAD [16,19,20], they are less accurate in predicting risk as a function of time or predicting molecular mechanisms driving disease progression. In contrast, data such as gene expression or protein expression reflect the environment and lifestyle of an individual. Such expression data can be measured from biosamples with current high-throughput molecular experiments, providing functional data which can be linked to the genotypes assessed in GWAS. Omics data involve the measurement of the entirety of a set of molecules such as mRNAs (transcriptomics) or proteins (proteomics) in a biological system (also referred to as *omics layers*). Functional omics data complement the genotyping data of a patient, and thus yields more detailed mechanistic insights into cellular processes, enabling, for instance, the identification of causal disease genes. Moreover, personalized gene regulatory networks may reveal genes or pathways involved in disease progression of a specific patient, and may thus allow for personalized development or prescription of drugs targeting particular genes or pathways. Specifically, novel drugs can then be developed to target an identified causal gene or pathway. Indeed, analyses involving omics data are promising enhancers of CVD risk prediction [21,22]. This further suggests an important role for multi-omics data, regulatory networks, and pathway analysis when predicting and understanding recurrent events.

Lately, a multitude of reviews have covered the topics of classical risk prediction models [23], artificial intelligence (AI) applied to recognize CVD in various forms of imaging data [24,25,26], or AI for the purpose of electrocardiogram-based prediction and drug discovery [26]. Additionally, reviews considering the molecular and genetic aspects of CVDs have been published in recent years, mostly focusing on describing the complex processes underlying disease [14,27,28]. These reviews consider incident CVD rather than recurrent events. In contrast, we here focus on the recent advances in risk prediction of recurrent cardiovascular events based on clinical and molecular data and discuss the use of AI as a tool for (1) predicting risk and (2) designing personalized treatment strategies. Because prediction models are often quantitatively evaluated with performance metrics, we provide a brief description of common metrics in Table 5. First, we summarize traditional risk prediction models based on clinical risk factors. We then transition to the molecular aspects of CVD and highlight the added value of biomarkers and networks to understand and predict patient risk. Finally, we describe how challenges of risk prediction can now be addressed with AI, and how explainable AI (XAI) may be an integral part of future transparent and reliable clinical decision systems which are based on both clinical and molecular data.

## 2. Results

### 2.1. Traditional Risk Scores Based on Clinical and Imaging Data

#### 2.1.1. Predicting the Risk of Incident CVD with Survival Models

The Framingham Risk Score (FRS) [2], Table 2 was the first modern risk prediction tool and is based on conventional clinical risk factors collected within the American Framingham heart study, Table 1. It was initially derived to predict the 10-year risk of CAD, but was later extended to predict a combination of CVD events: CAD, cerebrovascular disease, peripheral artery disease and heart failure [34]. Multiple studies have validated FRS on other cohorts [33,35,44]. The score has, for example, been found to overestimate the 10-year risk of CAD in a Japanese cohort [33], and the 10-year risk of CVD by 35% in a UK cohort [35].

The problem of FRS transferability has sparked the development of a multitude of risk scores using various regionally and ethnically diverse cohorts, as well as risk factor combinations. What all these scores have in common is the fact that the underlying historic data may overestimate event rates since better treatment modalities or improved environmental conditions may ameliorate risk over time. Table 1 and Table 2 summarize some well-known and established traditional risk scores, as well as the datasets from which they were derived. Examples of traditional risk scores include the guideline-recommended pooled European SCORE [30] (Systematic COronary Risk Evaluation), the pooled cohort equations recommended by the ACC/AHA (American College of Cardiology/American Heart Association) [31], the ASSIGN score [45] from the Scottish Intercollegiate Guidelines Network (SIGN), the UK QRISK scores [35,46], and the Japanese Suita score [33]. The QRISK score, inspired by the ASSIGN score, makes use of social deprivation and family history as additional risk factors. This, together with risk factors such as body mass index (BMI) and left ventricle hypertrophy increased performance, as measured by the area under the receiver-operating characteristics curve (AUC-ROC) on the UK dataset compared to ASSIGN and FRS. Later versions of the QRISK score are enhanced by additional risk factors related to atrial fibrillation, chronic kidney disease, migraine, and mental illness [46,47]. The NICE (National Institute for health and Care Excellence) guidelines of 2014, updated in 2016 [48], specifically recommend the QRISK2 score for estimating the 10-year risk of CVD. Inspired by QRISK2, the Suita score includes chronic kidney disease as an additional predictor, which improved the prediction of a 10-year risk for CAD on the Japanese cohort compared to FRS, according to the C-statistic (0.835 compared to 0.768).

#### 2.1.2. Predicting Recurrent Events with Survival Models

One of the first models for predicting recurrent events is the Global Registry of Acute Coronary Events (GRACE) score [4], which has been recommended by the European Society of Cardiology (ESC) since 2014 [49], Table 3a and Table 4a. It was designed to predict the 6-month risk of death and the combined risk of death or myocardial infarction (MI). According to the analysis, cardiac arrest upon admission and age were important predictors of death. The updated GRACE2.0 score [50] substituted uncommon attributes—such as the Killip class (to diuretic use within 24 h) and serum creatine concentration (to medical history of renal insufficiency)—in order to make the score more useful in practice.

Although the GRACE score has been successful, it is limited to predicting risk up to 6 months, while the risk of recurrent events can be elevated for several years after diagnosis. Models considering longer time-spans have therefore been developed in later years, including the models derived from the REduction of Atherothrombosis for Continued Health (REACH) registry [5], EuroAspire [3], TIMI [39] (Thrombolysis In Myocardial Infarction) and CONFIRM registry [9,43,51] (COronary CT angiography evaluatioN For clinical outcomes: An InteRnational Multicenter registry), Table 3b–e and Table 4b–e. The models derived from the REACH registry [5] were trained to predict recurrent cardiovascular events and death over a time-span of 20 months in patients with established CVD, Table 4b. The dataset contained data from 44 countries, including individuals from the Middle East. An external validation study, however, found that the model underestimated the 20-month risk of recurrent cardiovascular events in individuals from the United Arab Emirates, and showed an overall performance C-statistic of 0.63 (0.06 standard error) [52]. In general, the overall difference in risk between patients of different nationalities may be related to different genetics, different lifestyles, and environmental factors, or even a different quality of healthcare. Therefore, a model trained on a multi-national dataset may exhibit varying performance when evaluated on different nationalities. These multi-national models may thus benefit from recalibration to specific regional cohorts before being applied in practice [53].

The first score predicting the 10-year risk of recurrent cardiovascular events in patients with CVD was developed using the UCC-SMART (Secondary Manifestations of ARTerial disease) cohort [41] from the Utrecht university in the Netherlands, Table 3f and Table 4f. The dataset included clinical risk factors as well as carotid ultrasound findings. The SMART score [6], Table 4f, yielded a C-statistic of 0.683 (95% CI: [0.650, 0.717]), which slightly improved prediction compared to using the clinical risk factors (0.675, 95% CI: [0.642, 0.708]) or the carotid ultrasound findings (0.644, 95% CI: [0.609, 0.697]) alone [6]. The score is currently part of the ESC-recommended U-Prevent tool for predicting the 5- to 10-year risk of cardiovascular events when the patient has a history of CVD [53]. Still, recent studies have attempted to extend and improve the SMART score by including interventions as an end-point [54], and risk factors derived from CCTA images [8]. In the latter study [8], a 567-patient UCC-SMART sub-cohort with CCTA images was used to predict the 4-year risk of recurrent cardiovascular events or interventions. The additional CCTA-derived risk factors corresponded to coronary artery calcium, thoracic aortic calcium, and heart valve calcium. The findings showed that coronary artery calcium alone improved prediction accuracy from a C-statistic of 0.65 (95% CI: [0.59, 0.72]) without any calcification scores to 0.67 (95% CI: [0.61, 0.73]). By contrast, the thoracic and heart valve calcium did not improve risk prediction of recurrent events.

The added value of CCTA imaging risk factors has also been demonstrated by the CONFIRM risk score [43]. To build the risk score, the authors explored a mix of CCTA imaging and clinical risk factors to predict the 2-year risk of mortality for patients with suspected CAD, Table 3e and Table 4e. Based on the ability of CCTA risk factors to predict death (measured by the C-statistic of single Cox models), the final model consisted of a pre-trained score and two manually extracted CCTA risk factors: The number of nearby segments with more than 50% stenosis, and the number of nearby segments with calcified or mixed plaques. The CONFIRM score increased prediction performance in terms of the C-statistic relative to scores such as FRS (CONFIRM—0.682; FRS—0.639), showing the added value of CCTA-derived attributes for predicting mortality. Lately, this risk score has been enhanced with AI for the risk prediction task, as well as an extended list of CCTA and clinical risk factors [9,51] (see Section 2.3). Regardless, the CONFIRM score and the recent SMART score with CCTA risk factors [8] suggest that information such as coronary artery calcium or segments with calcified plaques from CCTA images improve risk prediction of recurrent cardiovascular events, interventions, and death. Although CCTA imaging is a non-invasive alternative for diagnosis of CAD, images are currently not available for all patients, which limits the practical use of these scores. Moreover, the image-based risk scores mentioned here may benefit from AI-based segmentation as opposed to manual extraction for identifying and extracting risk factors from the images.

A direct comparison between models based on the original papers is tedious due to the highly diverse set of end-points, time-spans, and cohort characteristics. Most of the models presented here, however, have an online risk calculator [4,6], or present the model explicitly in the original paper [5,43]. Model availability is not only the first step towards usability in a clinical setting, but also paves the way for more in-depth benchmarking studies.

Complementing clinical and imaging risk factors, recent research has shown that molecular data can further enhance risk prediction of incident CVD and recurrent events [55,56,57,58,59] as additional risk factors. Moreover, 40–50% of the risk of CAD is heritable [60], making genotype data in particular a potentially important aspect for predicting and characterizing the risk of recurrent cardiovascular events. In the next section, we will describe those molecular aspects in more detail.

### 2.2. Molecular Aspects of Risk Prediction of Cardiovascular Events

#### 2.2.1. Understanding CVD and Recurrent Events with Genotyping Data

Molecular data has been on the rise in recent years and is well-established in biomedical research. Figure 1 gives a brief overview on how molecular data can be used for the purpose of risk prediction of recurrent cardiovascular events. Genome-wide association studies (GWAS) have pinpointed hundreds of genetic loci and SNPs (Single Nucleotide Polymorphisms) contributing to the risk of developing CAD [14,16,61]. These SNPs also have value in predicting recurrent events as well as the efficacy of preventive treatments [17,62,63,64,65,66]. About a fifth of the CAD risk loci are located in the vicinity of genes related to CVD-relevant biological pathways, such as metabolism of lipoproteins or blood pressure [67]. The important roles of LDL cholesterol and hypertension for incident CVD events are well-established and therefore used in traditional models with clinical risk factors, Table 1. In line with this, current research has shown that lipoprotein(a) [68], or ceramides and phospholipids [69,70] in the blood pose good additional risk factors for risk prediction of recurrent events compared to the SMART and TRS2P scores in Chinese and Scandinavian CAD patients, respectively.

Studies based on genotype and conventional risk factor data from the GENIUS-CHD (GENetIcs of sUbSequent Coronary Heart Disease) consortium, Table 3g, showed that genetic variants associated with CAD may not necessarily be associated with recurrent events [71,72]. The known risk locus *9p21* [17,73], for example, did not show a strong association with recurrent acute CAD events [71]. This was demonstrated by the odds ratios from an age- and sex-adjusted meta-analysis of 49 studies, including almost 100,000 patients. In addition to this, the known CVD risk determinant *Factor V Leiden* was not associated with an increased risk of recurrent events, interventions or death in patients with CAD at baseline, as quantified by Cox hazard ratios [72]. The authors used 25 studies from GENIUS-CHD with a Factor-V Leiden genotype and conventional risk factors.

A disease-associated genetic locus, however, does not act in isolation, but rather through interactions with other genes, forming regulatory networks. So-called key-driver-genes have prominent positions in such networks as they, to some extent, determine the biological activity of a network [74,75]. Genetic variants modulating expression of key-driver-genes (eQTLs) come with a higher probability of affecting disease prevalence [28,76]. Unravelling the complex molecular architecture of gene regulatory networks giving rise to CAD is key to understanding disease progression, but may also be useful for predicting events. In general, some interactions between loci are assumed to be non-additive [18], necessitating the use of non-linear approaches such as AI models. However, so far, additive models have performed well in prediction tasks. For instance, additive models such as Polygenic Risk Scores (PRSs) (Figure 1) have so far been most successful to predict incident CVD [14,16,20,77,78], as well as recurrent events [79]. PRSs combine multiple genetic loci and are based on the fact that a combination of multiple variants of small effects can result in a larger risk for developing diseases like CAD, compared to specific rare monogenic mutations. The first genome-wide PRS for predicting CAD, among other diseases, was presented by Khera et al. in 2018 [16]. The results of this study have since then sparked new interest in PRS as a tool for predicting the genetic component of disease risk. Notably, a typical PRS consists of a two-step approach involving univariate variable selection through GWAS and subsequent multi-variate model fitting. The success of the PRS then depends on these two steps. In a recent study, a LASSO (Least Absolute Shrinkage and Selection Operator)-based PRS was developed. This approach allowed for estimation and variable selection simultaneously, thus merging the two steps into one [80]. The PRS was successfully applied to UK biobank [81] data, demonstrating scalability as well as predictive performance in accord with other established PRSs.

Although CAD PRS has proven valuable as an additional risk factor in traditional survival models such as FRS for predicting CAD [55,82,83], the added value of CAD PRS for predicting recurrent events is debated. Early studies concluded that genetic scores of incident CAD (i.e., not trained to predict recurrent events) may not improve risk prediction of recurrent events [84], while more recent studies found that genetic scores of incident CAD can indeed lead to an improved prediction performance of recurrent events [85]. In the former study [84], 5742 patients from the UCC-SMART dataset were used to train two Cox regression models for predicting the 10-year risk of recurrent major cardiovascular events. The first model used the predictors of the original SMART model, while the second combined these predictors with a genetic score of incident CAD, including 30 variants associated with CAD. The added genetic score did not increase prediction performance, as measured by the C-statistic. By contrast, a genetic risk score based on 79 CAD risk variants was recently shown to slightly improve predictability of recurrent cardiovascular events within 5 years in a cohort of 1,667 patients with ACS at baseline. The CAD PRS alone yielded a C-statistic of 0.581 (95% CI: [0.566, 0.596]), while the model using only clinical data obtained 0.686 (95% CI: [0.630, 0.658]). Compared to this, the combination of clinical data and PRS gave a slightly increased C-statistic of 0.698 (95% CI: [0.640, 0.668]) [85]. Another study investigated associations between incident-CAD PRS and cardiovascular or fatal events, as well as conventional risk factors, in individuals with and without CAD at baseline [79]. Using 408,480 individuals from the UK biobank and genome-wide association studies (GWAS) summary data from the CARDIoGRAMplusC4D consortium, the analysis showed a larger association between high PRS and greater risk of cardiovascular events in healthy individuals compared to those with CAD at baseline. In contrast to the healthy controls, patients with CAD at baseline further displayed an inverse association between the CAD PRS and death or ischemic stroke [79]. The results in these studies [79,84,85] together with the studies on the GENIUS-CHD dataset [71,72] suggest that the CAD PRS may not necessarily be an optimal predictor of recurrent event risk. Instead, a potential improvement for prediction of recurrent events may be to directly build a PRS of recurrent events [65].

A PRS may capture the genetic risk for developing CVD or recurrent events, but it does not fully explain the molecular disease etiology of CVD. Correlations between SNPs, for example, make it difficult to pinpoint causal genes. Moreover, risk contributors such as environmental and lifestyle factors cannot be explained with genomics data alone. Therefore, since both environment and genetics influence the levels of transcripts, proteins, and metabolites, molecular data encode their combined effect on the state of a cell. Analysis of these data facilitates exposure of new aspects, including the underlying disease mechanism, and may allow for identification of new predictive biomarkers [56,86,87]. High levels of the high-sensitivity C-reactive protein (CRP), for example, are associated with a higher 10-year risk for recurrent MI or death in CAD and ACS patients who underwent percutaneous coronary intervention (PCI) [56]. Moreover, increased levels of the monocyte chemoattractant protein-1 (MCP-1) have been suggested to increase the risk of recurrent events in patients with CAD [87,88]. Hence, functional multi-omics data may provide important information that might be lacking in clinical risk factors and genotype data.

#### 2.2.2. Integration of (Multi-)Omics Data

The possibility to generate readouts from different cellular information layers enables investigation of the molecular causes of a disease. Identified diagnostic biomarkers, for example, make it possible to guide treatment decisions and improve risk prediction. When collectively studied in a systems biology context, multi-omics have the potential to elucidate complex molecular mechanisms and pathways driving a disease [27,89]. This information can then be used to propose novel prognostic and therapeutic targets.

Overall, quantitative molecular data aid researchers in the transition from genetic and clinical data to mechanistic studies. To this end, genotype information may be correlated with mRNA, protein, or metabolite levels to identify disease-associated variants affecting those traits (quantitative trait loci (QTL), Figure 1) [75,90,91,92,93]. The genotype of a genetic variant can, for example, be correlated with the mRNA expression levels of genes across a cohort of individuals to determine expression quantitative trait loci (eQTL).

In the context of CVD, studies identified significant differences between the mRNA levels obtained from the left ventricular myocardium of patients with dilated cardiomyopathy and non-diseased controls [93]. Here, the authors further integrated differentially expressed genes with genotyping information and published GWAS loci, yielding numerous functional candidate genes relevant to cardiac phenotypes. Integrative approaches, where the molecular data are combined with clinical data for diagnostic or prognostic purposes, have shown especially promising results in the past [56,86,87]. In the following paragraphs, we will describe how disease-specific biomarkers can be derived from patient-level (multi-)omics data. Moreover, we highlight the use of molecular interaction networks to elucidate disease mechanisms.

**Biomarker discovery.**In general, biomarker discovery requires at least two sets of biosamples: One from healthy individuals (controls) and one from diseased individuals (cases, for example, diagnosed CVD). The goal is to obtain molecules which allow prognosis or diagnosis of a particular disease through distinctive concentrations. Prognostic markers are found by assessing molecular features at baseline prior to developing the disease or an event. In a straightforward case, significant differences between disease carriers and healthy individuals can be identified directly from quantitative molecular readouts of, for example, mRNA, protein or metabolite levels. The top distinguishing markers can then be found by applying a specific *p*-value and fold-change cutoff. To better understand the biology involved, these markers can be further scrutinized in pathway analyses—including gene ontology (GO, [94]) or Kyoto Encyclopedia of Genes and Genomes (KEGG, [95]) pathway enrichment analyses—as visualized in Figure 1. This allows to pinpoint disease-specific biological processes. Although markers derived from these computational analyses have a huge potential for diagnosis and prognosis tasks, they also require proper experimental and clinical validation before reaching routine clinical practice [96].

In a recent study, miRNA levels in whole-blood samples from a 437-patient cohort were collected from patients undergoing cardiac catheterization [97]. These data were used in a differential expression analysis to identify miRNAs differently expressed between patient groups. The cohort consisted of CAD patients with or without recurrent events (mean follow-up 1.5 years), as well as healthy controls without angiographic CAD. The differential expression analysis identified distinct miRNA profiles for individuals at risk of recurrent myocardial infarction. Interestingly, they were able to distinguish individuals on standard medical therapy exhibiting recurrent events from such individuals without recurrent events on the basis of miRNA profiles, but not using clinical attributes. This indicates a large prognostic potential and an added predictive value of miRNA profiles in risk prediction of recurrent cardiovascular events.

In another study [98], the authors performed a differential expression analysis between the proteomes of 85 CAD patients exhibiting recurrent events in a 5-year follow-up versus 85 sex and age matched CAD patients without recurrent events in this time period. Biosamples from blood serum were taken at baseline in light of the prospective Indian Atherosclerosis Research Study (IARS). The analysis identified β-defensin-128 and histatin-3 as potential biomarker predictors of recurrent CAD events. These biomarkers together with conventional risk factors improved risk prediction of recurrent cardiovascular events compared to the basic model with only conventional risk factors (C-statistic of 0.8 compared to 0.677). This further shows the huge potential of using omics, and specifically proteomics, in clinical settings.

**Network inference.** Although differential expression analysis is a rather routine approach to obtain a list of biomarker candidates for understanding molecular disease processes, analyses involving molecular networks facilitate identification of more robust biomarkers and causal mechanisms [75,99,100,101,102,103]. By reverse-engineering disease-specific regulatory networks or pathways, for instance in a multi-omics setting, it is possible to trace the effects of a disease-associated genetic variant throughout the cell [104] (Figure 1). This could lead to a better understanding of disease mechanisms. Moreover, identification of network modules allows to pinpoint clusters of genes which are most relevant to a disease. This bears a large drugability potential by revealing genes that are neighboring CAD candidate genes in relevant target tissues [75,105].

A recent study investigated gene expression microarray data from CAD patients without a history of MI, patients with MI but without heart failure, and patients with MI and heart failure (46, 34, and 30 cases, respectively) [102]. After detecting differentially expressed genes and long-non-coding RNAs (lncRNAs) between the three groups, the authors applied GO and KEGG pathway enrichment analysis to annotate the identified genes with their biological context. By applying weighted gene co-expression network analysis (WGCNA) to infer disease-relevant regulatory networks, they, for instance, found FASN and DGKZ together with their co-expressed lncRNAs as novel genes associated with CAD progression.

Another interesting approach is the direct inference of differential gene regulatory networks for cases and controls. One study for example proposed a novel strategy to infer group-specific networks using curated prior knowledge on protein-protein-interactions (PPI) in a cancer context [101]. The authors were able to derive a prioritized list of genes from their networks for more accurate survival prediction on independent datasets. Although in a different disease context (cancer), this framework could potentially be adapted for prediction of recurrent cardiovascular events.

The work presented in [106] highlights another application of gene regulatory networks. In their work, the authors inferred 28 CAD-related regulatory networks in diverse tissues around a set of eSNPs (**e**xpression **S**ingle-**N**ucleotide **P**olymorphisms: Genetic variants associated with gene expression levels). Through this, the authors identified novel genetic risk loci for CAD, which accounted for approximately 10% of heritability of CAD in addition to the 22% estimated by current GWAS studies. These new risk loci may be useful in the task of risk prediction by complementing current GWAS loci.

In another study, Franzén et al. utilized STAGE/STARNET genotype and gene expression data to investigate tissue-specific gene regulatory networks in CAD related tissues [103]. A step-wise approach was used to integrate cardiometabolic GWAS loci with tissue-specific expression quantitative trait loci (eQTL). The results were then fed into a causal inference pipeline to derive the gene regulatory networks. In particular, the results demonstrated that tissue-specific regulation of the *PCSK9* gene in visceral abdominal fat tissue, but not in other tissues, leads to an increased risk of elevated LDL/HDL ratios.

A different approach was implemented in a study by Sumathipala et al. [107]. The study curated a miRNA-protein-disease interaction network using several resources, such as miRTarBase [108], OMIM (Online Mendelian Inheritance in Man, OMIM®. McKusick-Nathans Institute of Genetic Medicine, Johns Hopkins University (Baltimore, MD, USA), 30 July 2021. World Wide Web URL: https://omim.org/ accessed on 29 August 2021) [109] and GWAS databases through DisGeNet [110]. The curated multi-omics network with miRNA–protein, protein–protein and protein–disease interactions was used to predict novel miRNA–disease interactions and to identify disease–disease relationships as well as disease subtypes via a novel network diffusion method [107]. Their findings suggest that specific miRNAs could identify disease subtypes, implying that miRNA biomarkers can increase patient stratification accuracy and, potentially, risk prediction. Moreover, the authors were able to predict differentially expressed miRNAs based on network information which could enable drug target discovery. Because miRNAs are known to affect CAD phenotypes [111], this work could be an important step towards improving clinical CAD models.

Although the methodological approaches mentioned here show great promise, most of them are not specifically applied to understand recurrent cardiovascular events. Added value could therefore be gained by exploring these approaches in such a context.

**Pathway analysis.** The goal of pathway analysis is to identify pathways or biological processes associated with CVD or recurrent events. The analysis may reveal known pathways and processes, such as cell adhesion, immune response, apoptosis, blood coagulation and inflammation [98,106]. Finding novel genes or pathways, on the other hand, may implicate potential therapeutic targets. In contrast to de-novo network and biomarker inference, one can rely on established pathway knowledge and known disease-relevant molecules and networks (Figure 1). For instance, if we observe many small but highly coordinated changes of gene expression in many genes of the same pathway, it is possible to estimate the combined effect of these genes. Moreover, one can identify disease association on a pathway level, which might be neglected when only analyzing individual genes. In fact, many of the studies referred to above which involve network inference rely on such knowledge to verify their biomarker candidates through enrichment analyses [97,102,106].

Overall, a large number of publicly available resources has been created for this purpose. Most of these resources—such as BioGrid [112], STRING [113], KEGG [95], and ConsensusPathDB [114]—focus on providing information about associations between molecular entities. Some, for example KEGG, also highlight associations with phenotypes or diseases, and provide a broader view of the biological context of genes. For a more complete overview on potential pathway resources, Bader et al. compiled a list of over 700 pathway databases in their Pathguide web resource (http://pathguide.org/, 30 July 2021) [115].

In a recent study [116], the authors inferred novel biomarker candidates with information provided by the GO, disease ontology (DO, [117]) and STRING databases, rather than validating the findings using GO or KEGG. They first curated a list of 608 potentially CAD-related genes from the CADgene database [118] and then followed a step-wise procedure to filter the list of genes and construct a protein-protein-interaction network. The network analysis pinpointed the single gene (EGFR) which is part of the ErbB signaling pathway. Specifically, increased levels of EGFR could lead to a 3-fold increase in CAD risk. The findings were validated via Western blotting in a CAD case-and-control cohort (*N* = 342/342).

With the advent of AI, new models have been developed for deriving polygenic risk scores, estimating pathway enrichment, and predicting risk based on molecular biomarkers or network structures. So far, most network-based methods rely on only a few omics layers or do not extend to a setting involving clinical and imaging data. Therefore, sophisticated AI may be useful for integrating all these data in novel prediction models. The next section will consider recent efforts where AI has been employed for predicting risk of CVD and recurrent events.

### 2.3. Exploiting Artificial Intelligence for Risk Prediction of Cardiovascular Events

Artificial intelligence (AI), including deep learning, has gained momentum in recent years and has been successfully applied to a wide range of medical applications [119,120,121,122,123,124]; a hot topic of many reviews [25,26,125]. In contrast to the traditional survival models, AI models can be trained to recognize complex and nonlinear patterns given a large set of data. Similar to Cox regression models, parameterized AI models may facilitate personalized treatments by assessing the patient-specific risk. AI models, however, often operate in a discrete fashion, predicting occurrence of an event, or risk at a specific time point, while traditional survival models predict the risk over years and are able to handle censored data.

Here, we will focus on the added value of AI models for predicting cardiovascular events. Although most contemporary research has focused on tailoring AI for risk prediction of CVD [44,126,127,128], we will also cover some recent models considering recurrent events [9,129]. First, however, we will provide a brief introduction of common terminology and AI models.

#### 2.3.1. Brief Introduction to AI

A machine learning, or AI, model is roughly either an *unsupervised* or a *supervised* method. Unsupervised methods are used to explore patterns in the data, for example by grouping observations according to some similarity criterion (i.e., cluster the data), without knowing about any groups beforehand. While unsupervised methods may be of great use to identify subgroups of patients with potentially different risk profiles, we will focus on supervised methods in this section. Supervised AI methods rely on labeled data: Each sample is paired with a label. For example, an individual which could be described by clinical values or molecular measurements may be labeled as “healthy” or “diagnosed with CVD”.

Figure 2A visualizes a typical AI workflow. Each patient is first described by a common set of features. A feature is a particular variable or patient attribute. In CVD risk prediction, a feature may correspond to a risk factor, molecular expression level, or smoking status, and so forth. The data are then divided into three sets: Training, validation, and test. During the *learning* or *training* phase, the parameters of the model are optimized based on an objective function which compares labels predicted by the model to observed (known) labels. Ideally, the final model should yield predictions that are as similar to the observed labels as possible, without compromising model generalizability, that is, avoid overfitting the model on the training data. Most AI models have non-trainable parameters, so called hyperparameters. To avoid overfitting, such parameters may be selected in a data-driven fashion with cross-validation. The model performance is then evaluated on the validation set according to a performance metric, Figure 2. The model with the hyperparameters yielding the best validation-set performance is selected as the final model and evaluated on the independent test set. A well-performing model can later be used to obtain predictions of single patients which were not part of the original dataset. Based on such predictions, the medical practitioner can assess the need for medications or additional follow-up visits.

Methods trained to recognize discrete class labels (e.g., healthy/diseased) are called *classifiers*, while methods trained on continuous labels (e.g., BMI or time to event) are referred to as *regressors*. AI models come in different flavors, including tree-based methods such as random forests [130] or XGBoost (eXtreme Gradient Boosting tree) [131], and various forms of neural networks [125,132] as examples. Tree-based methods typically consist of decision trees, which are made up of nodes. At each node of a decision tree, new branches are formed by splitting the training samples along a single feature. The split at a node is done to decrease the label heterogeneity of samples as much as possible at the resulting branches. Both random forest and XGBoost are ensemble methods consisting of multiple decision tree classifiers. Such methods avoid overfitting by voting or averaging across the different classifiers. Two other common methods are the *k*-Nearest-Neighbour (*k*NN) and Naive Bayes classifiers. In *k*NN, a patient’s risk status is predicted as the most common class label among the *k* samples which are closest to the patient in feature (patient attribute) space. Naive Bayes, on the other hand, is a model derived from probability theory. It assumes that all features are independent, conditionally on the class labels. Using this assumption, a sample (patient) can be associated with the most probable class label.

Neural networks attempt to mimic the activation patterns of connected neurons in a brain responding to input data [133]. The simplest form of a neural network is the so-called perceptron (Figure 2B). The perceptron is a binary classifier which returns a weighted sum of the input layer ”neurons” parsed through an activation function. The activation function can for example be a simple linear mapping or, in the case of a binary classifier, a logistic sigmoid function. As such, the perceptron is simply another way to describe logistic regression.

By adding additional layers (*hidden layers*) with their own weights and activation functions between the input and output layer, the perceptron is extended to a multi-layer perceptron—the vanilla form of neural networks (Figure 2C). The extra hidden layers with their activation functions is one way to introduce the recognized complexity and non-linearity of neural networks [132]. The information of a sample, or patient input data, is then propagated from the input to the output layer via a number of hidden layers; a so-called *feed-forward pass*. The opposite flow of information (output-to-input layer) is called *back-propagation*. Back-propagation is especially useful for training the network by passing the objective function gradient with respect to the weights from the final output back to the first input layer in order to minimize prediction error. Furthermore, back-propagation can be used for explaining the network predictions (see Section 2.4).

In addition to the multi-layer perceptron, neural networks exist in other forms including recurrent neural networks which are able to model sequence data, autoencoders which can be used for dimensionality reduction, and Convolutional Neural Networks (CNNs) [132]. CNNs were originally developed for recognizing and segmenting objects in imaging data. Inspired by image processing, the weights are constrained to identical values across space and can be viewed as filters which are convolved with the input image at each layer. The constraint effectively reduces the number of parameters compared to fully connected neural networks. This allows more hidden layers, and thus more nonlinearity, to be introduced. In medical research, CNNs have been successfully applied to imaging datasets, for example to facilitate identification of malignant white blood cells [121,122], automatic cardiac and aortic image segmentation [11,123,134], and scoring of coronary artery calcium [135]. The use for AI in cardiovascular image analysis has been thoroughly reviewed elsewhere [136].

In general, neural networks are flexible in terms of architecture and can therefore be tailored to specific tasks. CNNs have for instance been successfully applied to 1D genome sequence data for predicting the functional impact of non-coding SNP-variants on epigenetic features such as transcription binding sites and DNA accessibility [137,138,139]. Consequently, CNNs may be also well-suited for the task of risk prediction of disease or disease progression based on genotype data. In a recent study, a CNN was specifically designed for genotype data and was shown to improve prediction accuracy of whether or not an individual will develop amyotrophic lateral sclerosis (ALS) compared to simpler AI models [140].

#### 2.3.2. Utilizing Clinical and Imaging Data in AI Risk Prediction

Most studies so far have investigated the potential of AI for predicting incident disease rather than recurrent events, and typically make use of clinical risk factors. For example, an optimized AI pipeline—in terms of data imputation scheme, feature selection, and prediction model—was used to predict CVD in patients using 473 clinical data features from 423,604 patients from the UK biobank [44]. The so-called *autoprognosis* framework specifically selected this pipeline from a large pool of methods using Bayesian optimization. The selected and trained pipeline achieved an AUC-ROC of 0.774 (95% CI: [0.768, 0.780]), and thus outperformed both a Cox PH (proportional hazard) model fitted on the UK Biobank cohort using the conventional risk factors of the Framingham score (AUC-ROC 0.734 (95% CI: [0.729, 0.739]), Table 1) and a Cox PH model trained on the full set of attributes (AUC-ROC 0.758 (95% CI: [0.753, 0.763])), Table 6a.

In a study on prediction of presence of CAD, the performance of a CNN framework was compared to basic AI models, such as logistic regression, support-vector machine (SVM), and random forest [141], Table 6b. The models were trained on roughly 37,000 individuals described by 36 features from clinical data as well as demographic, laboratory and examination data. Although the findings could be strengthened statistically with for example confidence intervals, the results indicated that a CNN may perform similar to an SVM and random forest in terms of AUC-ROC and recall (0.768/0.773 compared to 0.774/0.776 and 0.7644/0.763), but slightly higher in terms of specificity (0.818 compared to 0.779 and 0.761). In another project, the added value of AI was investigated for the purpose of predicting the 5-year risk of CVD in a multi-ethnic cohort from Northern California [126], Table 6c. The ACC/AHA pooled cohort equations (PCE) [31] (Table 2) were compared to logistic regression, random forest, XGBoost and gradient boosted machine (GBM). Most AI models performed at least as well as the ACC/AHA pooled cohort equations in a patient-reduced dataset, Table 2. The three best models, the GBM, logistic regression with LASSO ($L_{1}$) regularization, and XGBoost, reached an AUC-ROC of 0.779 (95% CI: [0.760, 790]), 0.784 [0.765, 0.802] and 0.784 [0.766, 0.803] on the reduced dataset, while the ACC/AHA PCE reached 0.775 [0.755, 0.794] and logistic regression with $L_{2}$-regularization 0.749 [0.729, 0.770]. Although the simpler AI models did not necessarily improve performance, they allowed for a wider range of values in the input data and missing values than the ACC/AHA PCE. This allowed further improvement of the models in terms of AUC-ROC when trained on the full dataset, Table 6c.

In contrast to primary event prediction, another study derived three AI models (called the PRAISE models) for predicting the 1-year occurrence of recurrent MI, bleeding and death in patients with acute coronary syndrome (ACS) [129], Table 6d. Each model was constructed by selecting the best-performing classifier from a pool of four classifiers (adaptive boosting, naive Bayes, *k*NN, random forest). Model selection was performed using a validation set comprising 20% of the dataset. The models were trained on 25 features obtained from pooled international cohorts with clinical data of patients from Europe, Asia, North America, and South America. The final three models (adaptive boosting) achieved an average accuracy of 0.81–0.92 on the training dataset, as well as on the external (European) validation dataset) [129], Table 6.

The traditional CONFIRM model has lately been enhanced using a logit-boost model trained on a subset of approximately 10,000 patients from the CONFIRM dataset. The specific subset included extended CCTA imaging and clinical risk factors and longer follow-up time than the original model. The logit-boost model, which was trained to predict the 5-year risk of death in patients with suspected CAD, achieved an AUC-ROC of 0.79 (95% CI: [0.77, 0.81]). Thus, it outperformed traditional models such as the FRS (0.61 [0.59, 0.64]) and three CCTA-based scores (AUC-ROC 95% CI in the span [0.60, 0.66]) [9], Table 6e.

AI should in theory be able to model nonlinear data and therefore be superior to linear methods, such as the Cox model, for learning complex patterns. It can, however, be computationally expensive and sometimes tedious to select and train a model. The PRAISE models [129], together with the project by Ward et al. [126], suggest that ensemble models perform better than simpler AI models—including logistic regression, decision trees, *k*NN and naive Bayes—when predicting cardiovascular events and death, Table 6c–e. This was also suggested by the findings in a recent study predicting CVD in patients with type-2 diabetes [128]. The study, which was based on 172 CVD patients and 172 healthy controls, compared six models, namely naive Bayes, decision tree, random forest, SVM, logistic regression and *k*NN classifiers. Among these, random forest achieved the highest AUC-ROC (0.83 compared to 0.75–0.81 of the other models) [128]. In contrast, Dutta et al. found that the adaptive boosting classifier may perform similar to logistic regression, while the random forest and SVM had similar performance compared to the more advanced CNN when predicting the presence of CAD [141], Table 6b. The diverse results with respect to model performances may in part be attributed to insufficient choices of hyperparameters, thus highlighting some difficulties in finding optimal AI models. In general, simple models require less choices of hyperparameters and architectures, but are limited in complexity and flexibility.

In a clinical cohort, the follow-up data of some patients may be lost over time. This could be the case if no event happened within the time-frame of the study or if a patient drops out of a study by, for example, moving to a different region before exhibiting an event. The data are then *right-censored* at the time when the follow-up was lost. This type of data is difficult to directly make use of in basic AI models. Some AI models, however, allow for incorporation of survival models into their framework. Decision trees, and thus also random forest, can for example be adapted to right-censored data and be applied for risk prediction of cardiovascular events [142,143]. Moreover, the linear parametric function of the traditional Cox model can be replaced by a nonlinear function—for example, a neural network—which can improve the predictive performance compared to the original linear model [144,145]. Recently, an autoencoder neural network coupled with Cox regression was applied to sequences of cardiac magnetic resonance images to predict survival of 302 patients with pulmonary hypertension at baseline [11]. The model, which was trained on features from the CNN-segmented images, demonstrated an improved C-statistic (0.75, 95% CI: [0.70, 0.79]), compared to a traditional Cox regression model on clinical and imaging risk factors (0.64, 95% CI: [0.57, 0.70]), Table 6f.

#### 2.3.3. Utilizing Molecular Data in AI Models

The use for AI to enhance CAD polygenic risk scores (PRSs) is an intriguing research question. A CAD PRS based on random forest preprocessing was recently developed on the GerMIFS I-V and LURIC datasets, Table 6g [20]. The dataset initially contained ∼2.8 million SNPs after quality control, which were reduced to ∼98,000 by removing highly correlated SNPs. The variants were then filtered down to ∼50,000 using the importance of SNPs from a trained random forest prior to computing the PRS (see Section 2.4 on explainable AI). Only SNPs with nonzero importance were used for predicting the genetic risk of CAD. Interestingly, an additive PRS model on the filtered set of SNPs proved superior to AI models such as random forest, naive Bayes, SVM, and XGBoost, Table 6. This indicated that a linear additive PRS model may be enough to predict CAD and that increasing model complexity with AI may be of little advantage. Lately, this PRS has been benchmarked against the earlier genome-wide PRS by Khera et al. [16] on the MONICA/KORA dataset, showing similar performances [83]. In addition, both these PRSs increased performance of CAD risk prediction when added to conventional risk factors, demonstrating the utility of genetics in risk prediction [83].

Networks based on co-expression have traditionally been inferred solely from gene expression data and simple correlation measures. This, however, often yields highly connected networks owing to spurious correlations. Improvements of these traditional methods are therefore more restrictive in terms of the number of edges in the networks. Restrictions are enforced through regularization-based approaches involving partial correlations [146,147]. Other extensions aim at combining data from multiple omics layers. This way, the resulting regulatory networks span multiple cellular information layers [148,149,150]. Currently, new avenues for network inference are being explored in disease contexts. Graph neural networks employ deep learning to derive molecular network structures and disease-associated genes [151]. These neural networks can then be used for inference and prediction tasks. In prediction tasks, information about expression levels and molecular interactions are used to prioritize disease genes and identify patient sub-types, for example, patients at low or high risk of recurrent events.

With graph-CNNs, it is now possible to use the molecular networks directly as input to the risk prediction models [152]. The novel approach in [152] integrated multi-omics data to pinpoint new omics signatures and biomarkers for disease using a combination of neural networks and a View Correlation Discovery Network (VCDN). This method was successfully applied in several disease contexts, including Alzheimer’s disease and different types of cancers. In addition, graph-CNNs based on molecular data have been successful in predicting cancer genes [153,154] and cancer subtypes [155]. However, work is still needed to fully migrate this work into the field of CVD and recurrent events.

Multi-view neural networks have recently been proposed for analyses involving multi-omics data, including risk prediction [89]. This is carried out using a neural network with multiple input branches, such that each view corresponds to a specific type of omics data. The individual views then complement each other, ultimately leading to more comprehensive descriptions of the complex biological system at hand. Specifically, multi-view approaches allow to infer interactions between variables both within and between individual views [89]. As an example, multi-view models have been shown to outperform single modality-based approaches in predicting which patients with local memory loss symptoms will develop Alzheimer’s disease (AD) [156]. Hence, these models may also be useful in the context of risk prediction of recurrent cardiovascular events, especially when different types of patient data are available such as clinical, multi-omics and imaging data.

### 2.4. Explaining Decisions Made by AI Models

Machine learning or artificial intelligence (AI) models are traditionally described as “black boxes” where the patterns in the input data giving rise to certain outputs are unknown. In other words, an algorithm may be able to achieve a near to perfect performance in a classification task, but it might not be possible for humans to understand the intrinsic factors driving the classification. The problem with lacking insight into model decisions appears when a model learns to predict accurately based on irrelevant features and thus generalizes poorly to other datasets. A recent study, for instance, reported that an accurate AI model trained to identify COVID-19 in chest radiographs actually failed to make use of the relevant information in the images [157]. Due to a consistent patient positioning during imaging, the model instead recognized COVID-19-positive patients based on their shoulder regions. Unfortunately, the black-box behaviour of AI models is especially attributed to non-linear and complex models which are particularly suitable for advanced risk prediction.

Explainable AI (XAI) is a collection of methods which reveal the decision patterns of these complex models [158]. XAI may unveil novel aspects of a dataset and the disease itself by disclosing the predictive *importance* (or *relevance*) of input features, such as risk factors, attributes and protein expression levels. Simple models—e.g., principal component analysis (PCA), logistic regression and the Cox regression model—provide feature importance directly by their model parameters. Another simple model is the decision tree, which reveals decision patterns in the tree structure (see also Section 2.3.1). The so-called Gini feature importance is weighted by how much the impurity (label heterogeneity) of the data are decreased by a node split [159]. Such feature analyses provide an average importance over all samples (patients). This can often be extended to multiple classes by training models to distinguish each class from the rest, yielding insights related to, for instance, specific groups of patients [160]. However, the real power of explainable AI for clinical purposes comes with methods allowing to explain decisions made on individual patients. Figure 3 shows the main idea of using XAI in clinical decision systems. After the AI model has performed a patient-specific prediction, the medical practitioner can ask why the patient was associated with a certain risk, thus linking the model output to a specific set of clinical, imaging, or molecular attributes. This not only reveals the relevant risk factors, but may also be used to provide insights to the molecular mechanisms underlying disease by exploiting pathway databases and estimating gene set enrichment. All this information can guide the clinician when deciding on a treatment strategy.

Moreover, by knowing the decision pattern, the clinician would be able to assess faithfulness of the prediction. For this reason, explainability is suggested to be an ethical requirement for future clinical decision systems [157]. Here, we introduce different explainable AI methods and give a brief overview of applications in medicine and analysis of molecular pathways.

#### 2.4.1. Model-Specific Relevance Explanations

Explainable AI represents a large branch within the machine learning community. Most XAI methods are designed for a specific AI model, with lots of efforts put to deciphering neural networks. Within the group of model-specific methods, we for instance find the Gini importance for random forest classifiers [159], interpretable local surrogates [161], which estimate importance by approximation with a simpler model, and methods based on the architectural and back-propagation properties of neural networks [158,162].

Gini importance has found successful applications in pathway analyses [99,143,163]. An advantage of using random forests as opposed to the traditional univariate statistical testing is the implicit incorporation of interaction information, such as gene–gene interactions. In a random forest-based pathway approach, one classifier or regressor is trained per pathway category. These approaches use genes as predictor variables and estimate variable importance for each gene and pathway category. The distribution of gene relevances are then statistically compared between the genes of a specific pathway category and a subset of the background genes to test for pathway enrichment [143,163]. A recent benchmark study [99] found that random forest-based approaches outperformed traditional statistical methods on simulated data. The Gini importance provides an average variable importance over the sample population, which may help to disclose overall molecular drivers of the disease. With patient-specific XAI, on the other hand, it would be possible to obtain information about the pathways driving disease in each individual and thus possible to decide which drugs might be effective to treat that patient.

A patient-specific XAI method is the layer-wise relevance propagation (LRP), which is specific to neural networks. LRP operates via simple back-propagation rules which distribute relevance between layers [158]. The approach is thus able to disentangle otherwise highly complex and non-linear networks one layer at a time. The Deep Taylor Decomposition (DTD) framework specifically applies a Taylor decomposition at each layer while propagating relevance from the output layer to the input layer [162]. DTD can be viewed as a theoretical framework which encompasses multiple back-propagation relevance methods, including LRP [158]. Although DTD and LRP are able to provide feature relevance on a patient-specific level, they can only be applied to specific models, and an overall implementation supporting all neural network topologies is still missing.

#### 2.4.2. Model-Agnostic Relevance Explanations

A model-agnostic framework does not rely on a specific functional form or gradient and can therefore be applied to different types of AI models. A popular method for inferring feature predictability in supervised models is the permutation method, which calculates the increased prediction error after permuting the feature data. Although this can be applied to any AI model, a notable problem concerns correlated features. Specifically, two important yet correlated features may not affect prediction performance if not both features’ data are permuted. The permutation method is simple to implement and use but does not yield patient-specific relevance explanations.

The Shapley Additive Explanations (SHAP) [164], based on the game-theoretical Shapley value [165], is both model-agnostic and patient-specific. Intuitively, the Shapley value considers players in a game and tries to estimate the contribution of a player to the game outcome. In a risk prediction context, a game could correspond to a patient, a player to a risk factor, and a game outcome to the risk of developing disease or events. The contribution of a player can then be obtained by comparing game outcomes in the presence and absence of the player. This difference can be computed on a per-game (patient-specific) basis, making SHAP interesting for clinical decision systems. SHAP was recently presented and developed for tree models, such as XGBoost [164]. The theoretical framework, however, can be extended to any other model, including neural networks.

#### 2.4.3. Clinical Applications of XAI

XAI in clinical applications is still in its infancy, but LRP has already proven useful for interpreting CNNs used for image-based identification of MS [166], malignant leukocytes [122], and Alzheimer’s disease [167]. Moreover, LRP has been adapted for graph-CNNs and used to identify and characterize molecular signatures related to cancer [153,154].

Concerning model-agnostic XAI, the permutation method was used to obtain the patient-averaged feature importance from the PRAISE models predicting recurrent events [129], Table 6d. The analysis showed that left-ventricle ejection fraction, age, hemoglobin level and diabetes were important for predicting recurrent cardiovascular events and death within 1 year of being diagnosed with ACS. Another example is found in the original SHAP paper [164]. The authors used the NHANES dataset of 14,407 patients, 79 features, including clinical, demographic and lab attributes, and 20 year follow up to predict mortality risk and applied SHAP to further analyze the data and trained models. Their analysis revealed previously unknown interactions between features, such as age and sex, for mortality prediction. The data specifically suggested that the largest difference in mortality risk between men and women is observed at the age of 60 years. The large difference was suggested to be a result of the high risk of cardiovascular mortality for men relative to women.

Altogether, XAI represents an intriguing avenue which, together with the flexibility of AI models, may greatly improve contemporary risk prediction models of recurrent cardiovascular events.

## 3. Discussion

Accurate risk prediction models are crucial for facilitating early prognosis and prevention of recurrent cardiovascular events, as well as for formulating personalized treatment strategies. Smart decision systems in clinical settings would therefore not only save human lives, but also reduce expenses related to follow up visits and interventions, alleviating this large burden on the individual as well as the society.

Given the recent studies highlighted in this review, risk prediction of recurrent cardiovascular events can be improved in two ways in the future: (1) by using better-quality datasets with both clinical and molecular attributes, and (2) by using advanced, yet explainable, artificial intelligence (AI) models. Patient care could be improved drastically by leveraging the data already gathered in routine clinical work and combining them with additional molecular data. Therefore, the rise of multi-omics data in clinical research and the increased digitization of medical facilities is especially promising. However, the prospects of recurrent-event risk prediction in the context of CVD require additional efforts where molecular and clinical data are brought together into a unified framework. Moreover, clinical data are not always available directly to researchers, and technical and data privacy issues need to be overcome to successfully use them in medical research.

A recent comprehensive benchmark of 11 AI models and 18 datasets showed that, in a cancer context, integration of multi-omics and clinical data in an AI framework can sometimes improve prediction of survival time compared to the traditional Cox model based on clinical attributes alone [86]. The added value of omics data was especially clear in models which weighed or treated the omics layers and clinical data differently in a block-like fashion. Novel AI models with flexible architectures may therefore be particularly advantageous when handling datasets with a mix of traditional clinical risk factors, imaging, and multi-omics data. However, the benchmark also showed that whether or not a specific AI model leveraging multi-omics data actually improved prediction compared to a Cox model trained on clinical data alone depended on the specific dataset at hand. In the context of recurrent cardiovascular event risk prediction, the PRAISE models based on adaptive boosting [129] displayed a high AUC-ROC (Table 6d) and good calibration, but were not validated or compared to traditional scores such as the GRACE [50] or TIMI [39] scores (Table 4a,d). Moreover, although the CONFIRM logit-boost model (Table 6e) outperformed traditional models trained on less attributes [9], it was not compared to a Cox model trained on the same attributes as the logit-boost model. Thus, more work is needed to properly benchmark the predictive ability of simpler AI models, ensemble models, neural networks and the Cox model when predicting recurrent cardiovascular events. A proper benchmark of the different traditional and AI models listed in this review would require one or, preferably, multiple datasets with multi-national individuals. The dataset(s) should be big enough to enable thorough statistical analysis of national- or event-specific model predictability. Moreover, they should not only contain the correct data of all risk factors used in each model, but also information about the different event outcomes of interest. Although most datasets can be used to evaluate a basic model, such as the Framingham risk score, they typically cannot be used to evaluate models with uncommon attributes—such as risk factors extracted from imaging data—due to missing data. Therefore, a completely unbiased comparison of these models on different datasets is difficult.

Although successful in predicting incident CAD, the use of known genetic CAD risk loci for predicting recurrent events is still under investigation. For instance, the application of polygenic risk scores (PRS) for CAD, while beneficial for predicting incident disease, does not necessarily enhance prediction of recurrent events. This issue could, however, be alleviated through systematic evaluation and application of state-of-the-art network and pathway modelling. Numerous resources containing both functional omics data and known interactions between molecular entities have been established for this purpose. Current developments in artificial intelligence and machine learning, such as graph neural networks, could advance recurrent risk prediction by properly utilizing these resources. As discussed in this review, novel biomarkers predicting risk could be implicated by combining clinical risk factors with disease-associated genetic variants, additional molecular data and with established knowledge about molecular interactions.

Finally, clinical decision systems developed in future studies should be able to explain their predictions on a patient-level basis. This would inform the medical practitioner about prediction reliability, and aid the formulation of a personalized treatment plan. In particular, explainable AI may reveal the functional disease mechanisms on a molecular level when multi-omics data together with pathway databases are included in the decision system framework. Being able to integrate the different layers and types of data via a comprehensive explanatory AI framework may thus render a more holistic view of the patient and their risk for recurrent cardiovascular events. New developments in the AI field need to take these factors into account in order to mature from research to clinical applications.

## Figures and Tables

**Figure 1 ijms-22-10291-f001:**
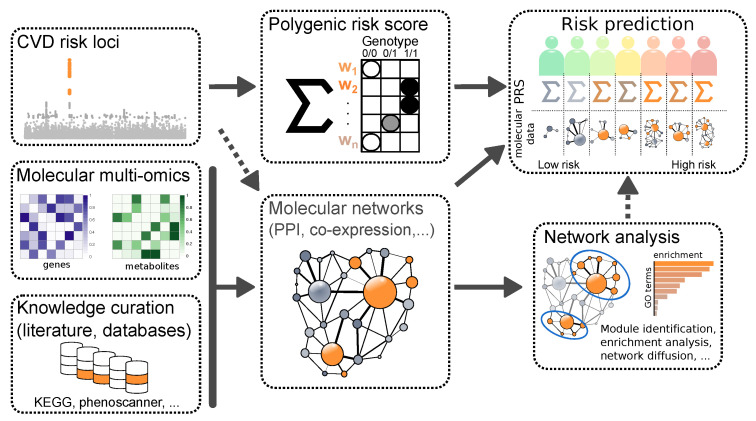
Overview of how molecular data can be used for understanding and predicting the risk of recurrent cardiovascular events. Genome-wide association studies (GWAS) can be used to identify CVD risk loci. Weights obtained from GWAS can be used to calculate a polygenic risk score. Moreover, the GWAS loci can be combined with multi-omics data and prior knowledge to construct regulatory networks. From these networks, it is possible to extract physiological pathways and network modules, as well as associate the level of activity of distinct network regions with high or low risk. The network information and polygenic risk score can be integrated together to improve risk prediction of recurrent cardiovascular events.

**Figure 2 ijms-22-10291-f002:**
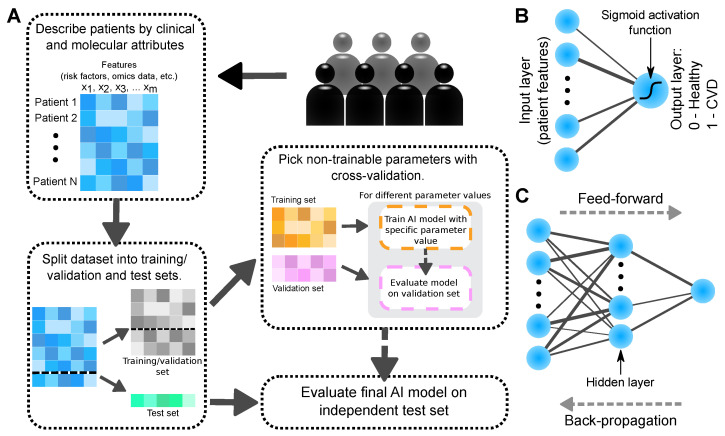
(**A**) Illustration of a typical AI workflow. Each patient is first described by the same set of numerical and/or categorical attributes (features), such as risk factors or gene expression levels. The data (patients) are then divided into training, validation and test sets. AI models with different values of non-trainable parameters (hyperparameters) are trained on the training set. The model performance is evaluated on the validation set according to some metric. A final model with the hyperparameters yielding the best validation-set performance is then evaluated on the independent test set. (**B**) Illustration of a perceptron neural network with a sigmoid activation function. (**C**) Illustration of a multi-layer neural network with one hidden layer. The arrows indicate direction of feed-forward and back-propagation passes.

**Figure 3 ijms-22-10291-f003:**
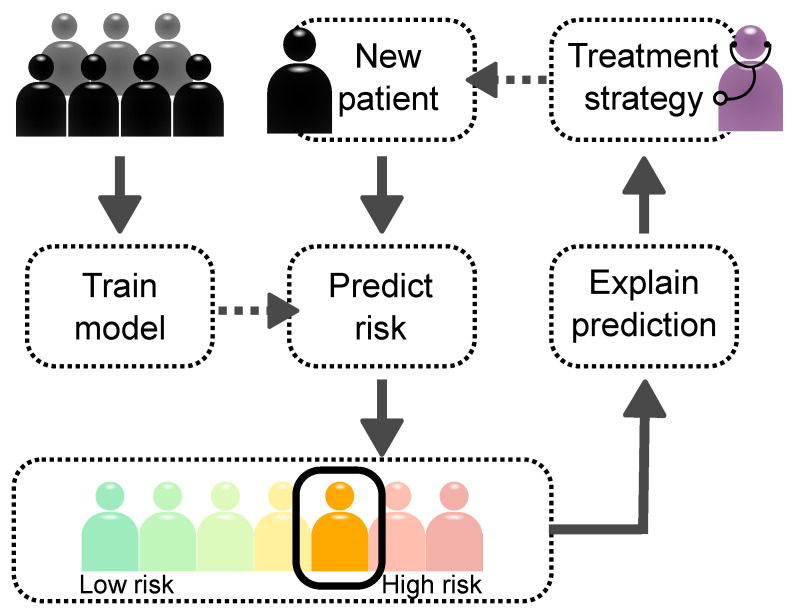
A typical clinical decision system with explainable AI. An AI model is first trained on a cohort containing for example clinical, imaging or multi-omics data. The trained model is then used to predict the risk for a patient to develop the disease or specific symptoms. Finally, the explainable AI provides information about the decision patterns, which helps the medical practitioner to assess faithfulness of the prediction and formulate a treatment strategy.

**Table 1 ijms-22-10291-t001:** Table of clinical datasets used to train traditional risk scores. LDL—Low-density lipid; HDL—High-density lipid; BMI—Body mass index; LV—Left ventricle; CKD—Chronic kidney disease.

Dataset	Cohort Size	Attributes	Follow Up	Ref.
		**FRS attributes:**		
		age, sex,		
American Framingham	∼15,000	diabetes,	12 years	[29]
heart study		LDL and HDL		
		cholesterol, smoking,		
		systolic blood pressure		
		Sex, smoking,		
SCORE pooled dataset	∼21,000	total cholestrol,	Average	[30]
(12 European cohorts)		tot. chol./HDL ratio,	13 years	
		systolic blood pressure		
ACC/AHA		FRS attributes, tot. chol.		
pooled	∼25,000	treated/untreated	≥12 years	[31]
cohort		systolic blood pressure		
		FRS attributes,		
QRESEARCH		social deprivation,		
(No diabetes or	∼10 million	family history,	17 years	[32]
CVD at baseline)		BMI, LV function,	(study length)	
		antihypertensive		
		agent treatment		
Suita dataset	∼5600	FRS attributes, CKD	11.8 years	[33]

**Table 2 ijms-22-10291-t002:** Table of traditional risk scores based on clinical datasets. MI—myocardial infarction; TIA—transient ischaemic attack; CAD—coronary artery disease; CVD—cardiovascular disease; PH—proportional hazards.

Risk Score	Dataset	Clinical Question	Method	Performance	Ref.
				0.733–0.841/	
Framingham	American	10-year risk;		0.769–0.847	
risk score	Framingham	CAD,	Cox PH	(C-statistic,	[34]
	heart study	CVD events		events average,	
				male/female)	
	SCORE	10-year risk;		0.71–0.84	
SCORE	pooled	CAD/CVD	Weibull	(AUC-ROC,	[30]
	dataset	mortality		EU countries)	
ACC/AHA	ACC/AHA	10-year risk;		0.713–0.818	
pooled cohort	pooled	atherosclerotic	Cox PH	(C-statistic,	[31]
equations	cohort	CVD events		sex/race average)	
		10-year risk;		0.7674/0.7879	
QRISK	QRESEARCH	CVD (MI, CAD,	Cox PH	(AUC-ROC,	[35]
		stroke, TIA)		male/female)	
Suita	Suita dataset	10-year risk;	Cox PH	0.835	[33]
		CAD		(C-statistic)	

**Table 3 ijms-22-10291-t003:** Table of clinical, genetic, and imaging datasets used to predict risk for recurrent events. CCTA—coronary computed tomography angiography; CVD—cardiovascular disease; CAD—coronary artery disease; CeVD—cerebrovascular disease; PAD—peripheral artery disease; ACS—acute coronary syndrome.

Dataset	Cohort Size	Type of Data	Baseline	Follow Up	Ref.
		Clinical			
(**a**) GRACE	∼102,000	risk	ACS	6 months	[36]
registry	(30 countries)	factors			
		Clinical risk	CVD		
(**b**) REACH	∼68,000	factors,	(CAD, CeVD,	1–2 years	[37]
registry	(44 countries)	demographics	PAD)		
		Clinical			
(**c**) EuroAspire	IV/V: ∼16,000	risk factors,	CAD	0.5–3 years	[38]
	(27 countries)	lifestyle			
		Clinical			
(**d**) TIMI	∼8600	risk	ACS	2.5 years	[39]
		factors		(median)	
		CCTA images,			
(**e**) CONFIRM	∼50,000	clinical	suspected	2.3 years	[40]
registry	(6 countries)	risk factors	CAD	(median)	
		Clinical risk			
		factors, carotid			
(**f**) UCC-SMART	∼13,000	ultrasound,	CVD	4.7 years	[41]
		(567 patients with CCTA images		(median)	
	∼186,000				
(**g**) GENIUS-CHD	(57 studies,	Genotype	CAD	1–15 years	[42]
	18 countries)				

**Table 4 ijms-22-10291-t004:** Table of clinical risk scores for predicting recurrent events. CVD—cardiovascular disease; MI—Myocardial infarction; PH—proportional hazards; CI—confidence interval.

Risk score	Dataset	Clinical Question	Method	Performance	Ref.
	GRACE registry	6-month risk;		0.7/0.82	
(**a**) GRACE	(43,810 patients,	death, or	Cox PH	(C-statistic,	[4]
	14 countries)	death/MI		death/death-MI)	
	REACH	20-month risk;		0.67 [0.66, 0.68]/	
	registry	CVD events,		0.75 [0.73, 0.77]	
(**b**) REACH	(49,689 patients,	cardiovasc.	Cox PH	(C-statistic,	[5]
	44 countries)	death		95% CI,	
				CVD/death)	
	EuroAspire	2-year risk;			
(**c**) EuroAspire	(IV/V,	CVD events	Weibull	0.67 [0.64, 0.70]	[3]
	27 countries,	or		(C-statistic,	
	12,484 patients)	interventions		95% CI)	
		3-year risk;			
(**d**) TRS2P	TIMI	Cardiovasc.	Cox PH	0.67 [0.65, 0.69]	[39]
	(8598 patients,	death, MI,		(C-statistic,	
	9 predictors)	stroke		95% CI)	
	CONFIRM	2-year risk;			
(**e**) CONFIRM	registry	death	Cox PH	0.682	[43]
	(20,300 patients)			(C-statistic)	
	UCC-SMART	10-year risk;		0.68 [0.64, 0.71]	
(**f**) SMART	(5788 patients,	CVD events	Cox PH	(C-statistic,	[6]
	14 predictors)			95% CI)	

**Table 5 ijms-22-10291-t005:** Description of performance metrics. Here, a “positive” label can, for example, correspond to “having CVD”, while a “negative” may correspond to “healthy”.

Metric	Description	Math. Definition
True positive	A positive sample correctly	TP
(TP)	predicted by the model.	
True negative	A negative sample correctly	TN
(TN)	predicted by the model.	
False positive	A sample wrongly classified as	FP
(FP)	positive by the model.	
False negative	A sample wrongly classified as	FN
(FN)	negative by the model.	
Precision	Fraction of true positives among	TP
	the predicted positives.	TP+FP
Recall	Fraction of positives that are	TP
(Sensitivity)	correctly predicted.	TP+FN
Specificity	Fraction of negatives that are	TN
	correctly predicted.	TN+FP
Accuracy	Fraction of correctly predicted	TN+TP
	positives and negatives.	TP+TN+FP+FN
ROC curve	A curve indicating performance	
(Receiver Operating	of a classifier. The *Y*-axis shows	R(s)
Characteristic)	recall and the *X*-axis shows	
	s = (1-specificity)	
AUC-ROC	Quantitative performance	
(Area Under the Curve	measure based on ROC curve.	
- ROC)	Ranges from 0 to 1, where 1	∫R(s)ds
	corresponds to perfect, and 0.5	
	to random, classification.	
C-statistic	Equivalent to AUC-ROC. Can	
	be used for censored data	
	(missing patient outcomes).	
	$r_{i}$—predicted risk of patient *i*	$\sum_{ij}\varepsilon_{i}1\left(r_{i}>r_{j}\right)1\left(t_{i}<t_{j}\right)$
	$t_{i}$—time to event, patient *i*	$\sum_{ij}\varepsilon_{i}1\left(t_{i}<t_{j}\right)$
	$\varepsilon_{i}\in \left\{0,\;1\right\}$ - whether (event)	
	information exists.	
	1(x)={1ifx,0otherwise}	
PR curve	Similar to ROC curve. *Y*-axis	
(Precision Recall)	shows precision and *X*-axis	P(r)
	recall (*r*).	
AUC-PR	Quantitative performance	
(AUC—Precision-Recall)	measure based on PR curve.	∫P(r)dr
	Alternative to AUC-ROC.	

**Table 6 ijms-22-10291-t006:** Performance of AI models for predicting primary or recurrent events compared to traditional survival models. CVD—cardiovascular disease; CAD—coronary artery disease; ACS—acute coronary syndrome; PH—proportional hazards; CI—confidence interval; NPHS—National Pulmonary Hypertension Service; EHR—Electronic health records; LR—Logistic regression; RF—Random forest; MLP—multi-layer perceptron; NB—Naive Bayes.

Risk Score/	Dataset	Clinical	Prediction	Comparison	Ref.
Method		Question	Performance	(Cox PH)	
(**a**) Auto-	UK Biobank	5 year-risk;	*AUC-ROC, 95% CI:*	*All attributes:*	
prognosis	(clinical data,	Fatal or	0.774	0.758	
framework	423,604	non-fatal	[0.768, 0.780]	[0.753, 0.763]	[44]
	patients)	CVD event		*FRS attributes:*	
				0.734	
				[0.729, 0.739]	
(**b**) CAD CNN	NHANES	Predict	*AUC-ROC:*	-	
**(1)** CNN	(clinial, lab,	presence	**(1)** 76.87		
**(2)** LR	demographic	of CAD	**(2)** 71.29		[141]
**(3)** SVM	data,		**(3)** 77.64		
**(4)** RF	37,079		**(4)** 76.24		
**(5)** AdaBoost	patients)		**(5)** 71.63		
**(6)** MLP			**(6)** 72.61		
(**c**) ASC AI	EHR	5-year risk;	*AUC-ROC, 95% CI:*	*ACC/AHA:*	
	(clinical data,	MI, stroke,	*(full/reduced)*	*(full/reduced)*	
**(1)** GBM	socioecomics	or fatal CAD	**(1)** 0.835 / 0.779	- /0.775	
	262,923/		[0.825, 0.846]/	[-, -]/	
	131,721		[0.760, 0.790]	[0.755, 0.794]	
**(2)** LR–$L_{1}$	patients)		**(2)** 0.784/0.825		
			[0.765, 0.802]/		
			[0.812, 0.839]		
**(3)** XGBoost			**(3)** 0.784/0.830		[126]
			[0.766, 0.803]/		
			[0.816, 0.843]		
**(4)** RF			**(4)** 0.773/0.831		
			[0.760,0.793]/		
			[0.820,0.842]		
**(5)** LR-$L_{2}$			**(5)** 0.749/0.808		
			[0.729,0.770]/		
			[0.795,0.820]		
(**d**) PRAISE	BleeMACS,	1-year risk;	*AUC-ROC, 95% CI:*	-	
(AdaBoost)	RENAMI	**(1)** Death,	**(1)** 0.82 [0.78, 0.85]		
	(clinical data,	**(2)** MI,	**(2)** 0.74 [0.70, 0.78]		[129]
	19,826	**(3)** bleeding	**(3)** 0.70 [0.66,0.75]		
	patients)	(ACS at baseline)			
(**e**) CONFIRM	CONFIRM	5-year risk;	*AUC-ROC, 95% CI:*	*FRS:*	
(logit-boost)	registry	Death	0.79	0.61	[9]
	(10,030 patients)	(suspected CAD	[0.77, 0.81]	[0.59, 0.64]	
		at baseline)			
**Method**		**Question**	**Performance**	**(Cox PH)**	
(**f**) 4D-survival	NPHS	Survival-times	*C-statistic, 95% CI:*	*Clinical/imaging*:	
(survival	(MR imaging,	*(pulmonary*	0.75	0.64	[11]
autoencoder)	clinical data,	*hypertension*	[0.70, 0.79]	[0.57, 0.70]	
	302 patients)	*at baseline)*			
(**g**) CAD PRS	GerMIFS I-V,	Genetic risk;	*AUC-ROC, 95% CI:*	-	
**(1)** PRS	LURIC	CAD	**(1)** 0.92 [0.90, 0.94]		
**(2)** SVM	(∼ 2.8M SNPs,		**(2)** 0.82 [0.80, 0.85]		[20]
**(3)** NB	15,709 patients)		**(3)** 0.82 [0.79, 0.84]		
**(4)** RF			**(4)** 0.75 [0.72, 0.78]		
**(5)** XGBoost			**(5)** 0.74 [0.71, 0.77]

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
