# Peer review of "Risk Prediction of Cardiovascular Events by Exploration of Molecular Data with Explainable Artificial Intelligence"

_ijms, 2021, doi:10.3390/ijms221910291_

Round 1
Reviewer 1 Report
The authors of this review article "Risk Prediction of Cardiovascular Events by exploration of Molecular data with Explainable Artificial intelligence" have provided a comprehensive literature analyses on the limitations of traditional risk prediction using clinical and imaging approaches and highlighted the possible benefits of integrating molecular data in improving prognostication/risk prediction of recurrent CAD (Coronary artery disease) related events. In addition, they provide a review of the emerging importance of using Artificial intelligence (AI) in integrating and analyses of multiple types data for better risk predictions in cardiovascular disease. Overall the review is well written and figures are appropriate. However, I suggest minor corrections to the text/tables as outlined below.
1. line 31 of the text the authors state "current treatment algorithms.." I suggest the use of 'protocol' or 'regimen' instead of 'algorithm' as the latter term is more appropriate in relation to automation and not human generated workflows as is the case in most clinical scenarios.
2. line 62-63 is not clear. Please rewrite.
3. For tables 2-6, please include appropriate references as a separate column in each table.
Reviewer 2 Report
The authors introduced the application of AI technique on the prediction of CVD. Overall, this is a well-written and well-organized review. I just have few comments for authors to further improve their work. The detailed comments are listed as follows:
1 The difference of this current review from the existing reviews should be mentioned in the introduction part.
2 Summary of exemplified CVD studies with deep learning approach should be included.
3 A main graphic abstract that will be invaluable to attract readers’ attention at first sight is needed.
4 The application of AI approach to the personalized outcome shall be supplemented.
5 Future prospects of CVD study shall be discussed at last.
